# Effects of Deoxynivalenol-Contaminated Diets on Productive, Morphological, and Physiological Indicators in Broiler Chickens

**DOI:** 10.3390/ani10101795

**Published:** 2020-10-02

**Authors:** Insaf Riahi, Virginie Marquis, Antonio J. Ramos, Joaquim Brufau, Enric Esteve-Garcia, Anna Maria Pérez-Vendrell

**Affiliations:** 1Institute of Agrifood Research and Technology (IRTA Mas Bové), Animal Nutrition Department, 43120 Constanti, Spain; insaf.riahi@irta.cat (I.R.); joaquim.brufau@irta.cat (J.B.); enric.esteve@irta.cat (E.E.-G.); 2Phileo by Lesaffre, 137 rue Gabriel Péri, 59700 Marcq en Baroeul, France; v.marquis@phileo.lesaffre.com; 3Applied Mycology Unit, Food Technology Department, University of Lleida, UTPV-XaRTA, Agrotecnio, Av.Rovira Roure 191, 25198 Lleida, Spain; antonio.ramos@udl.cat

**Keywords:** deoxynivalenol, broilers, guidance level, performance, organ weights, small intestine, blood biochemistry, welfare

## Abstract

**Simple Summary:**

The presence of mycotoxins in feed is a really significant problem worldwide; it leads to adverse effects on animals and great economic losses, especially in the monogastric industry. Deoxynivalenol (DON) is one of these mycotoxins that contaminates poultry feed and consequently has negative effects on this specie. Different concentrations of DON (5 and 15 mg/kg feed) were tested in broiler chickens. Results showed that high levels could adversely affect productive and welfare parameters; however, organ weights, morphological intestine indicators, and biochemical parameters were affected at low and high level of dietary DON. In general, even the low level of DON (5 mg/kg), which is the guidance level in complete poultry feed could affect the physiological status of birds.

**Abstract:**

The present study with 1-day-old male broilers (Ross 308) was conducted to evaluate the effects of deoxynivalenol (DON) at different levels (5 and 15 mg/kg feed) on growth performance, relative weight of organs, morphology of the small intestine, serum biochemistry, and welfare parameters of broiler chickens. Forty-five broiler chicks were randomly divided into three different experimental groups with five replicates each: (1) control group received a non-contaminated diet, (2) contaminated diet with 5 mg DON/kg of feed, and (3) contaminated diet with 15 mg DON/kg of feed for 42 days. Results showed that feed artificially contaminated with DON at guidance level (5 mg/kg diet) did not affect growth performance parameters. However, 15 mg/kg reduced body weight gain and altered feed efficiency. DON at two assayed levels significantly increased the absolute and relative weight of thymus and the relative weight of gizzard and decreased the absolute and the relative weight of the colon. Compared to controls, both doses affected small intestine morphometry parameters. In terms of biochemical indicators, DON at 5 mg/kg reduced the creatine kinase level and at 15 mg/kg DON reduced the cholesterol level. Furthermore, DON at 15 mg/kg induced more fear in broilers compared to broilers fed the guidance level. It was concluded that even the guidance level of DON did not affect the chickens’ performance. However, its toxic effect occurred in some organs and biochemical parameters.

## 1. Introduction

Deoxynivalenol (DON) is a secondary metabolite produced by *Fusarium* fungi. DON is the most widespread mycotoxin found in grains such as wheat, rye, barley, maize, oats, and their byproducts [1,2]. A recent survey, reported that DON is the most frequent contaminant of feedstuffs in Europe [3]. From 4311 samples evaluated, 63% were positive, with an average positive level of 0.6 mg/kg and a maximum of 40.7 mg/kg. For poultry feeding stuffs, the recommended maximum level for DON is set at 5 mg/kg [4], although lower doses of DON could cause adverse effects on performance [5,6]. On the contrary, other reports indicated that DON at higher than the recommended tolerance value did not cause adverse effect on zootechnical parameters. Body weight gain, feed intake, and feed conversion ratio of broiler chickens were not affected by the inclusion of naturally or artificially contaminated diets with DON at 10 mg/kg DON for 35 and 42 d of age [7,8,9,10]. Furthermore, an important list of investigations indicated that this toxic effect could be observed when birds fed concentration of DON greater than 15 mg/kg [11,12].This suggests that chickens may be relatively tolerant to this *Fusarium* toxin. However, it has been observed that DON-contaminated feed could affect organ weights, blood biochemical, and immunological parameters [13,14]. Interestingly, it was reported that broilers exposed to dietary DON had a higher stress index (heterophil to lymphocyte ratio) [15,16], meanwhile there are few studies regarding the effects of DON on stress of animals. The fear response is a welfare-related behavior as expressed by tonic immobility reaction and can provide more information on bird’s stress status [7,17]. However, lack of information was reported on the effect of DON feeding on this indicator.

The first purpose of this study herein was to test two doses of DON mycotoxin to better understand tolerable DON level in broilers feed, to explore potential damages a lower level might cause, and to see if, in the practice conditions, broilers could tolerate up 15 mg/kg of DON through evaluating some indicators of effect of toxicity such as productive, morphological and physiological parameters. Furthermore, due to the lack of information about DON effect on bird’s welfare, we hypothesized that both doses can affect the welfare stratus in the terms of fear response and footpad color.

## 2. Materials and Methods

### 2.1. Ethical Approval

All animal care procedures were approved by the Ethical Committee for Animal Experimentation of IRTA, in accordance with current regulations on the use and handling of experimental animals (Decree 214/97, Generalitat de Catalunya, Catalonia, Spain).

### 2.2. Experimental Design, Birds, and Diets

A total of forty-five 1-d-old male chicks (Ross 308) was randomly distributed in fifteen battery cages from 1 d to 4 d of age. Temperature in the first two days was 34 °C, then the temperature was gradually decreased 3 °C weekly to 21 °C until the end of the experiment. Twenty-four hours of light was provided during the first two days with a reduction to 18 h until 7 days and 14 h afterward. Chickens were fed a starter diet from 1 to 21 days and a grower diet from 22 to 42 days. Diets were formulated to meet or exceed broilers requirements according to [18]. Chickens of treatment 1 received a starter and grower basal diet composed of maize (54 and 59%), soybean meal (36 and 31%), soy oil (4.9 and 5.7%), monocalcium phosphate (1.42 and 1.30%), minerals, amino acids, and a premix with vitamins and minerals (0.49 and 0.44%). Chickens of treatments 2 and 3 received basal diet contaminated with DON at 5 mg/kg feed or 15 mg/kg feed, respectively. Each treatment had 5 replicated pens with each pen containing 3 birds. Feed in mash form and water were provided for *ad libitum* consumption.

### 2.3. Analyses

Representative feed samples for each group were analyzed for the content of dry matter, crude protein, gross energy, crude fiber, ether extract, crude ash, and sodium chloride [19] (Table 1). For the trial, DON was produced by growing in vitro *Fusarium* graminearum strain I159 on wheat, in accordance to the protocol described by [20] (ENVT, Toulouse, France), and was mixed into the experimental feed. The powdered culture material was included to obtain 5 or 15 mg DON/kg feed. DON levels were confirmed by liquid chromatography mass spectrometry. In the basal starter diet, DON concentration was 65 µg/kg feed, while in DON-contaminated starter diets, DON was 4760 and 14,390 µg/kg for treatments 2 and 3, respectively. The level of DON in the grower basal diet was 73 µg/kg feed, and 4650 and 15,120 µg/kg for treatment 2 and 3, respectively. HPLC analysis also confirmed that aflatoxin B1 (AFB1) was below the limit of detection (0.3 µg/kg). Zearalenone (ZEN), fumonisins (FBs) and ochratoxin A (OTA) were determined using Ridascreen^®^ and Elisa kits (R-Biopharm) following the manufacturer’s instructions. The limit of detection (1.75, 25, and 2.5 μg/kg for ZEN, FBs, and OTA, respectively) (Table 1).

### 2.4. Productive Parameters and Organ Weights

Chickens were tagged in their wing and weighed individually, and feed consumption for each pen was determined at 21 d, 35 d, and 42 d of age. Body weight gain and feed intake were calculated and feed conversion ratio was calculated as the ratio between feed intake and body weight gain for 21 d, 35 d, and 42 d. The incidence of mortality was recorded daily. On d 42, chickens were weighed individually, and 15 birds in each treatment were humanely euthanized according to IRTA ethics instructions. Proventriculus, pancreas, heart, gizzard, liver, kidneys, small intestine, cecum, colon, spleen, thymus, and bursa of Fabricius were excised and weighed. Organ weights were presented as direct measured and defined as absolute organ weights. Relative organ weights were expressed as a percentage of body weight.

### 2.5. Morphological and Histological Traits of Small Intestine

The gastrointestinal tract of chickens (12/treatment) euthanized at 42 d (from the duodenum to the ileocecal junction) was weighed after removal of the content by gentle squeezing. Absolute weight and length were determined and relative weight and density were calculated. Density of the intestines was calculated as the ratio between the absolute weight in grams and the length of the intestine in centimeters.

The jejunum was considered the segment between the end of the duodenal loop and Meckel’s diverticulum. Intestinal jejunum samples were taken close to the junction of Meckel’s diverticulum and then sampled and fixed in 10% neutral-buffered formalin solution for at least 48 h. Each tissue sample was dehydrated in serial alcohol baths, cleared in xylene, and embedded in paraffin wax using an automatic tissue processor system (Leica, TP 1020, Barcelona, Spain). Tissue blocks were mounted, sectioned at 4 µm, and stained with hematoxylin and eosin. Intestinal jejunum samples were observed with a light microscope (BHS, Olympus, Barcelona, Spain). Villus height and crypt depth were measured. Measurements were taken in 10 well-oriented villi and crypts from each intestinal section of each animal [21]. Villus height and crypt depth were measured using a linear ocular micrometer (Olympus, Microplanet, Barcelona, Spain). Results are expressed in µm. Villus: crypt ratio was calculated dividing villus height by crypt depth.

### 2.6. Blood Biochemistry

At d 42, blood samples (1 mL/bird, 3 birds/pen) were collected by cardiac puncture into non heparinized tubes for serum biochemistry. Serum samples were obtained by centrifugation at 978× *g* for 10 min. Total protein, albumin, aspartate aminotransferase (AST), alanine aminotransferase (ALT), cholesterol, alkaline phosphatase (ALP), gamma-glutamyltransferase (ᵞ-GT), glucose, lactate dehydrogenase (LDH), uric acid, and creatine kinase (CK) were determined using an automatic biochemical analyzer (Olympus AU5800, Beckman Coulter, Brea, CA, USA).

### 2.7. Fear Behavior and Leg Color

Fear levels (tonic immobility reaction) were determined according to the method described by [7,17]. Briefly, at d 28 and 35 of the trial, 15 birds per treatment were treated individually to determine the fear test. In a separate room, bird was placed on its back on a laboratory table and was restrained for 45 s on its sternum by experimenter hand. Then, the experimenter removed his hand gradually. If the bird remained immobile for 20 s. a stopwatch was started to record the time until the bird righted itself. The latency until righting was recorded is defined as the duration of tonic immobility reaction. If the bird righted itself in less than 20 s, the tonic immobility reaction had not been induced, and another induction was started. In the case that the bird did not right itself over 10 min, a maximum score of 600 s was given for the tonic immobility duration.

At the end of the trial the footpad color of 3 chickens per replicate was determined by means of a Minolta CR-300 with CIE Lab color system: L* corresponds to lightness, a* to green–red chromaticity, and b* to blue–yellow chromaticity.

### 2.8. Statistical Analysis

Data were analyzed as a completely randomized design and were showed as means ± SEM. Each cage was considered the experimental unit. Data were analyzed by one way analysis of variance (ANOVA) using the General Linear Model Procedure of SAS software (SAS 9.4, SAS Institute, Cary, NC, USA) to examine the effect of different treatments. Orthogonal polynomials were used to determine linear and quadratic dose responses. To test the normal distribution of data, Kol–Mogrov–Smirnov test was used. All statements of differences were considered significant at *p* ≤ 0.05 and trends were considered at 0.05 < *p* < 0.10.

## 3. Results

### 3.1. Growth Performance and Organ Weight

The results of productive parameters and mortality are listed in (Table 2). After 42 days of trial, DON presence at 5 mg/kg did not affect these parameters compared with the control group (*p* > 0.05). However, a tendency of decrease in a linear way (*p* = 0.08) was observed in broilers fed 15 mg/kg feed on body weight gain (BWG) which was 6% lower than broilers fed the control diet. In addition, DON at 15 mg/kg altered the feed conversion ratio (*p* = 0.03) in a linear way. No significant effects of dietary treatments were observed on mortality. The results of absolute and relative organ weights are presented in (Table 3). Compared to control group, DON treatments elevated the absolute weight of thymus (*p* = 0.0002) and decreased the absolute weight of colon (*p* = 0.01). Moreover, an increase in the relative weight of gizzard (*p* = 0.006) and thymus (*p* < 0.001) and a reduction in the relative weight of colon (*p* = 0.03) was observed after feeding different levels of DON (5 and 15 mg/kg) to chickens aged 42 d.

### 3.2. Morphological Traits of Small Intestine

Density of the small intestine was defined as the ratio of the absolute weight of small intestine to the length. A reduction in the absolute weight (*p* = 0.05) and a reduction trend of the relative weight of the small intestine and small intestine (a trend) (*p* = 0.09) was observed after exposure to DON in broilers. Furthermore, the small intestine was longer (*p* = 0.01) for birds fed dietary DON compared to control chickens resulting in a lower density (*p* = 0.001) for birds fed DON (Table 4). No significant effect was observed on villus height, crypt depth, and ratio of villus height to crypt depth (Table 5).

### 3.3. Blood Biochemistry

There was no effect of DON on blood biochemical values (*p* > 0.05), except for a decline in creatine kinase at 5 mg/kg DON in quadratic way (*p* = 0.04) and cholesterol level at 15 mg/kg DON (*p* = 0.004) (Table 6).

### 3.4. Fear Behavior and Leg Color

The effects of experimental treatments on tonic immobility reaction and footpad color are shown in (Table 7). Birds fed DON at 15 mg/kg showed longer tonic immobility duration than birds fed 5 mg/kg at 35 d (*p* = 0.05). Furthermore, no marked differences among the diets groups were detected for the footpad color.

## 4. Discussion

DON mycotoxin is frequently detected in poultry feed and chickens are considered relatively tolerant to DON in terms of growth and performance parameters [8], although several studies show highly variable effect of DON on these parameters. Growth performance parameters of broiler chickens were not affected by DON guidance value (5 mg/kg) inclusion in experimental groups in present study. Likewise, lack of effect of DON at 5 mg/kg in feed on body weight gain, feed intake and feed conversion ratio was reported by researchers [22,23]. Even at 10 mg/kg tested in broiler diets, body weight, bodyweight gain, feed intake, and feed conversion ratio were not adversely affected for 35 d [7,8] and for 42 d [9,10]. These results observed might be suggestive of adaptation of birds to mycotoxins over time and that poultry are relatively tolerant to DON mycotoxicosis compared with other species especially pigs due to the differences in DON absorption, distribution, metabolism, and elimination (ADME) [24]. Osselaere et al. [25] found that the absolute oral bioavailability in broilers chickens was accounting for only 19.3%, and therefore contributing to the poor absorption of DON. Furthermore, they reported that DON is characterized by its high clearance and rapid elimination. The tolerance of broilers to DON could also be explained by the extensive metabolization of DON in DON-3 sulphate; this phase II metabolite is much less toxic than DON itself [26]. Interestingly, DON at 15 mg/kg tended to decrease the BWG and altered the feed conversion ratio at the final of the experiment. This finding is consistent with the finding of [13]. The differences occurred by the presence of 15 mg/kg DON in feed could be justified by the mode of action of this toxin that is principally the inhibition of protein synthesis at the elongation or termination steps [27].

Previous studies suggested that the organ weights could be influenced by mycotoxins [28]. The most susceptible tissues to trichothecenes mycotoxicosis, including DON, are those with high protein turnover rates, such as the immune system (bone marrow, lymph nodes, spleen, and thymus), the liver, the intestinal mucosa, and the small intestine [27]. Consequently, it was expected that DON might affect immune organs (such as thymus, spleen, and bursa of Fabricius), liver, and small intestine weights as organ targets. In the current study, DON treatments increased significantly the relative weight of gizzard. Similarly, an increase in the relative weight of this organ after DON exposure (up to 16 mg/kg feed) was observed in other studies with broiler chickens [9,17]. The increase in relative weight of gizzard may be related to a difference in the density of the diet, or may be a consequence of irritation of the upper gastrointestinal tract [29]. Birds exposed to highly DON-contaminated diets (82.8 mg/kg) for 27 d had small erosions in the gizzard mucosa [30]. The changes observed in weights of thymus and colon were not mentioned in previous reports. However, the relative weight of thymus of broilers exposed to DON (from 1 to 12.20 mg/kg for 5 w) did not change significantly [31,32]. Similarly, the relative weight of colon was unchanged in broilers fed dietary DON at different levels 1, 5, and 10 mg/kg for different experimental periods (3, 5, and 6 w) [10,22,31]. The increase in relative weight of thymus induced by DON treatments showed in this study was expected as thymus is a lymphoid organ and with high protein turnover. From this current study, a reduction in absolute weight and a trend of the reduction in relative weight of small intestine were noted. Relative weight of small intestine of broilers fed 5 mg/kg DON for 21 d decreased [22]. Similarly, Yunus et al. [33] tested two levels of DON in broiler chickens, low level (1.68 mg/kg feed) and high level (12.20 mg/kg feed), and reported a reduction in the relative weight of small intestine segments (duodenum and jejunum). This reduction may be strongly related with other morphological changes observed when birds were fed DON such as the decrease in villus height [33]. Thus, the higher relative weights of gizzard and thymus and the reduction in the relative weights of colon and small intestine after DON intoxication suggests that DON caused a non-direct effect on chickens that may include either enlargement or atrophy of the internal organs, probably due to irritation or cell damage, or it is an indicative of the animal’s necessary immune responses to dietary contaminations [28].

The gastrointestinal tract (GIT) is the first barrier against ingested chemicals, feed contaminants, and natural toxins [34]. The intestinal epithelium cells can be exposed to high concentrations of DON following ingestion of contaminated diets [35]. The morphology of the intestine could give some information on gastrointestinal development [36]. Therefore, as intestinal morphology indicators of toxicity of DON, parameters included in the absorption process were evaluated such as the length of small intestine, villus height, and crypt depth. Birds fed DON in the current investigation had longer small intestines and lower density than birds fed the control diet. Consistent with results found by [33], the length of duodenum and jejunum increased, and therefore, the density of small intestine decreased significantly in broilers fed DON low level (1.68 mg/kg feed) and DON high level (12.20 mg/kg feed). We speculate that this may be explained by the decrease in villus height induced by dietary DON (numerical decrease in this study), which could be accompanied by lower electrophysiological properties, which resulted in low absorption of glucose in the small intestine [9]. Furthermore, Yunus et al. [33] indicated that with the increase in length and the density decrease in the small intestine, higher absorption of DON, reduction in the absorption of nutrients, and therefore, a decrease in performance was expected.

It was reported that DON could reduce villus height, which was explained by villus contraction and resulted in nutrient transport and utilization impairment and, therefore, could impair zootechnical parameters [23,37]. However, in the current study, this decrease did not reach statistical significance, probably due to the difference of number of samples between treatments. It has been reported that diets containing DON at different levels (2, 5, and 10 mg/kg feed) for 16 weeks have no significant effects on villus height of jejunum and ileum of Taiwan country chickens [5]. In addition to that, crypt depth of broiler chickens was not affected by DON treatments (1 and 5 mg/kg) during 35 d [31]. Similar results regarding crypt depth were observed when broiler chickens were fed with 18 mg DON/kg feed for 21 d [16]. However, decreased villus height of small intestine of broilers exposed to DON mycotoxin has been described in other studies [23,33].

From the present study, it is clear that high DON levels decreased serum cholesterol level. This change could be related to possible damage in liver function induced by DON mycotoxin. Similarly, Kubena et al. [38] observed a reduction in the serum cholesterol level of White Leghorn chickens fed 18 mg/kg of DON-contaminated grains for 28 d. The authors explained this decrease to liver involvement and a shift of concentrations from the blood to the liver. In their meta-analysis, Andretta et al. [39] reported that broilers fed challenged mycotoxins (T2, FBs, DON, OTA, and ZEN) showed lower total cholesterol compared to negative control birds and that this reduction was about 14%. In contrast, other authors have described increases in cholesterol levels. Thus, higher cholesterol levels in broilers fed DON (10 mg/kg) for 35 d was found by [8]. The authors suggested also that this increase may be due to liver or kidney function damage or stress. Additionally, DON-contaminated diet at 5 mg/kg decreased the creatine kinase level, and this reduction could reflect the rates of the loss of this enzyme from the circulation [40] or cell damage with leakage of the contents into the blood [39]. Andretta et al. [39] indicated that broilers fed challenged mycotoxins (T2, FBs, DON, OTA, and ZEN) had lower CK than the negative control birds (−27%).

Few studies suggested that the stressful effect of *Fusarium* mycotoxins included DON appeared through the alteration of the brain regional neurochemistry. For example, the inclusion of contaminated grains with *Fusarium* mycotoxin in the diet of broiler chickens increased concentrations of 5-hydroxytryptamine (5 HT) in the pons and in the cortex [41]. Knowing that the 5-HT system takes place in the regulation of fear, we suggested that DON could affect the fear level in broilers. Therefore, stress-related behavior in the current study was evaluated through evaluating the duration of tonic immobility at week 4 and 5, which was prolonged in birds fed a high DON level compared with birds fed a low DON level. These results indicated that chickens fed a higher level of DON had a higher level of fear and probably reinforce the poor performance induced at this level.

## 5. Conclusions

In conclusion, broiler chickens could tolerate dietary contaminated DON in terms of growth at the guidance value but not at 15 mg/kg. However, the effect of DON is apparent from 5 mg/kg to 15 mg/kg on organ weights such as the gizzard, thymus, and colon and on the small intestine morphology. In addition, the effect of both doses was apparent on the physiological and behavior status of the birds through impacting the blood cholesterol, creatine kinase level, and fear response. The changes obtained on parameters evaluated appear as indirect response to the dietary DON and give a better understanding of the DON-tolerable level in poultry feeding. Further studies should evaluate relevant biomarkers of toxicity of DON related to the health and welfare of chickens. 

## Figures and Tables

**Table 1 animals-10-01795-t001:** Analyzed composition and mycotoxin contamination of the experimental diet.

Item	Control Group	DON ^1^ Group (5 mg/kg)	DON Group (15 mg/kg)
Broiler starter			
Dry matter (%)	88.9	88.9	89.0
Crude protein (%)	21.8	21.3	21.5
Gross energy (Kcal/kg)	4094	4147	4170
Crude fiber (%)	2.40	2.20	2.48
Ether extract (%)	6.75	7.13	7.24
Ash (%)	5.56	5.65	5.65
Sodium chloride (%)	0.30	0.32	0.32
Mycotoxins (µg/kg)			
DON	65	4760	14,390
ZEN	<1.75	84.4	242
FBs	142	257	216
OTA	0.94	0.90	1.21
AFB1	<0.3	<0.3	<0.3
Broiler grower			
Dry matter (%)	89.1	89.1	89.1
Crude protein (%)	19.6	19.8	19.6
Gross energy (cal/g)	4186	4208	4240
Crude fiber (%)	2.24	2.37	2.48
Ether extract (%)	7.84	7.94	8.02
Ash (%)	4.98	4.97	4.96
Sodium chloride (%)	0.31	0.35	0.33
Mycotoxins (µg/kg)			
DON	73	4650	15,120
ZEN	<1.75	85.9	259
FBs	225	216	275
OTA	1.59	1.11	1.10
AFB1	<0.3	<0.3	<0.3

^1^ DON = deoxynivalenol; ZEN = zearalenone; FBs = fumonisins; OTA = ochratoxin; AFB1 = aflatoxin B1; limit of detection of DON, ZEN, FBs, OTA, and AFB1: 50, 1.75, 25, 0.5, 0.3 µg/kg, respectively.

**Table 2 animals-10-01795-t002:** Effects of DON-contaminated feed (5 and 15 mg/kg) on growth performance of broiler chickens.

Dietary Treatment ^1^							
Item	Control	DON (5 mg/kg)	DON (15 mg/kg)	SEM	*p*-Value	Linear	Quadratic
BWG (g/d/bird)							
d 0 to 21	29.1	30.7	30.4	1.98	0.83	0.70	0.65
d 0 to 35	50.2	50.7	48.5	1.73	0.67	0.46	0.60
d 0 to 42	58.5 ^a^	57.4 ^a,b^	54.7 ^b^	1.39	0.22	0.08	0.94
Feed intake (g/day/bird)							
d 0 to 21	42.4	43.0	40.4	1.28	0.38	0.24	0.43
d 0 to 35	72.4	72.4	69.7	2.20	0.64	0.38	0.73
d 0 to 42	85.3	86.9	88.0	2.82	0.80	0.53	0.85
Feed conversion ratio (g:g)							
d 0 to 21	1.46	1.41	1.40	0.05	0.73	0.53	0.65
d 0 to 35	1.44	1.42	1.43	0.01	0.94	0.77	0.86
d 0 to 42	1.45 ^b^	1.51 ^ab^	1.55 ^a^	0.02	0.09	0.03	0.49
Mortality (%)							
d 0 to 42	13.3	19.9	13.3	8.16	0.80	0.90	0.52

^1^ DON, deoxynivalenol; SEM, standard error of mean (*n* = 12); ^a,b^: means values with different superscripts with the same row differ (*p* ≤ 0.05).

**Table 3 animals-10-01795-t003:** Effects of DON-contaminated feed (5 and 15 mg/kg) on organ weights of broiler chickens.

Dietary Treatment ^1^							
Item	Control	DON (5 mg/kg)	DON (15 mg/kg)	SEM	*p*-Value	Linear	Quadratic
Gizzard, g	39.3	42.6	39.4	1.41	0.19	0.76	0.07
Liver, g	53.3	56.5	52.6	2.55	0.53	0.70	0.31
Kidneys, g	13.4	14.0	13.3	0.58	0.71	0.81	0.43
Colon, g	2.89 ^a^	2.04 ^b^	2.12 ^b^	0.21	0.01	0.03	0.03
Thymus, g	9.58 ^b^	16.5 ^a^	14.2 ^a^	1.10	0.0002	0.02	0.0003
Bursa of Fabricius, g	6.44	5.87	5.15	0.53	0.26	0.10	0.83
Gizzard, %	1.46 ^b^	1.65 ^a^	1.62 ^a^	0.04	0.006	0.04	0.01
Liver, %	1.99	2.19	2.08	0.08	0.25	0.61	0.11
Kidneys, %	0.51	0.54	0.53	0.02	0.58	0.60	0.36
Colon, %	0.11 ^a^	0.08 ^b^	0.08 ^b^	0.008	0.03	0.09	0.04
Thymus, %	0.33 ^b^	0.63 ^a^	0.57 ^a^	0.03	<0.0001	0.001	<0.0001
Bursa of Fabricius, %	0.24 ^a^	0.22 ^b^	0.19 ^b^	0.02	0.22	0.08	0.95

^1^ DON, deoxynivalenol; SEM, standard error of mean (*n* = 12); ^a,b^: means values with different superscripts with the same row differ (*p* ≤ 0.05).

**Table 4 animals-10-01795-t004:** Effects of DON-contaminated feed (5 and 15 mg/kg) on small intestine morphology of broiler chickens.

Dietary Treatment ^1^							
Item	Control	DON (5 mg/kg)	DON (15 mg/kg)	SEM	*p*-Value	Linear	Quadratic
Small intestine, g	67.7 ^a^	59.9 ^b^	59.3 ^b^	2.34	0.02	0.02	0.10
Small intestine, %	2.57 ^a^	2.34 ^b^	2.34 ^b^	0.08	0.09	0.09	0.09
Length, cm	192 ^b^	206 ^a^	208 ^a^	4.30	0.01	0.01	0.10
Density, (g/cm)	0.34 ^a^	0.29 ^b^	0.28 ^b^	0.01	0.001	0.002	0.02

^1^ DON, deoxynivalenol; SEM, standard error of mean (*n* = 12); ^a,b^: means values with different superscripts with the same row differ (*p* ≤ 0.05).

**Table 5 animals-10-01795-t005:** Effects of DON-contaminated feed (5 and 15 mg/kg) on small intestine histology of broiler chickens.

Dietary Treatment ^1^						
Item	Control	DON (5 mg/kg)	DON (15 mg/kg)	*p*-Value	Linear	Quadratic
Villus height (µm)	956 ± 47.1	921 ± 36.3	928 ± 82.4	0.90	0.79	0.76
Crypt depth (µm)	128 ± 10.0	102 ± 11.3	99 ± 11.0	0.12	0.11	0.31
Villus height-to-crypt depth ratio	7.84 ± 0.53	9.46 ± 1.02	9.74 ± 0.90	0.14	0.11	0.36

^1^ DON, deoxynivalenol; values are least-square means ± SEM (standard error of the means); (*n* = 12, 5, 7 by treatments), respectively.

**Table 6 animals-10-01795-t006:** Effects of DON-contaminated feed (5 and 15 mg/kg) on blood biochemistry of broiler chickens.

Dietary Treatment ^1^							
Item	Control	DON (5 mg/kg)	DON (15 mg/kg)	SEM	*p*-Value	Linear	Quadratic
Total protein (g/L)	29.7	29.6	30.0	0.60	0.92	0.74	0.81
Albumin (g/L)	10.1	10.0	9.84	0.21	0.56	0.29	0.90
AST (U/L)	332	285	314	18.6	0.24	0.71	0.10
ALT (U/L)	2.46	2.16	2.23	0.21	0.57	0.52	0.41
Cholesterol (mmol/L)	3.83 ^a^	3.55 ^ab^	3.29 ^b^	0.13	0.004	0.001	0.47
ALP (U/L)	5807	4849	4893	647	0.48	0.37	0.43
ᵞ-GT (U/L)	23.7	22.5	20.8	1.50	0.36	0.16	0.90
Glucose (mmol/L)	13.9	13.8	14.0	0.16	0.74	0.72	0.50
LDH (U/L)	3894	3689	3200	416	0.32	0.13	0.95
Uric acid (mmol/L)	335	298	279	22.5	0.21	0.10	0.52
CK (U/L)	9532 ^a^	4412 ^b^	7527 ^ab^	1610	0.10	0.63	0.04

^1^ DON, deoxynivalenol; SEM, standard error of mean (*n* = 12); ^a,b^: means values with different superscripts with the same row differ (*p* ≤ 0.05).

**Table 7 animals-10-01795-t007:** Effects of DON-contaminated feed (5 and 15 mg/kg) on welfare parameters of broiler chickens.

Dietary Treatment ^1^							
Item	Control	DON (5 mg/kg)	DON (15 mg/kg)	SEM	*p*-Value	Linear	Quadratic
Fear behavior							
Tonic immobility duration (s)							
28 d	132	103	118	36.8	0.85	0.86	0.59
35 d	169 ^ab^	85.7 ^b^	246 ^a^	43.7	0.05	0.09	0.06
Number of inductions							
28 d	1.61	1.50	1.84	0.31	0.72	0.52	0.62
35 d	1.38	1.25	1.53	0.20	0.61	0.49	0.47
Footpad color							
L	78.0	77.8	77.6	0.51	0.87	0.62	0.88
a	3.24	2.87	3.13	0.34	0.74	0.94	0.45
b	30.6	33.7	31.9	1.54	0.35	0.70	0.16

^1^ DON, deoxynivalenol; SEM, standard error of mean (*n* = 12); ^a,b^: means values with different superscripts with the same row differ (*p* ≤ 0.05).

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
