# Peer review of "Effects of Deoxynivalenol-Contaminated Diets on Productive, Morphological, and Physiological Indicators in Broiler Chickens"

_animals, 2020, doi:10.3390/ani10101795_

Round 1
Reviewer 1 Report
In the study entitled as “Effects of Deoxynivalenol Contaminated Diets on Productive, Morphological and Physiological Indicators in Broiler Chickens”, authors investigated the effect of Deoxynivalenol on various parameters. The study is worthy and of significance in the poultry industry.
Some corrections needed to improve the clarity of the manuscript.
-Line no. 17; Deoxynivalenol (DON) … insert the abbreviation.
-Line no. 22; “(5mg/kg)” There is a space between the number and unit Please check out throughout the manuscript.
-Could you please elaborate, why male broilers were selected? What about female?
-Is there any possible explanation of the correlation between fear and DON? It could be due to nervous system damage or any other reasons. If possible, write 2-3 sentences in discussion section.
-In some places, comma is missing. Please check out the whole manuscript.
-Delete this sentence “Thereafter, evaluate the efficacies of DON-detoxifying products on those biomarkers specified.”
Author Response
Point 1: Line no. 17; Deoxynivalenol (DON) … insert the abbreviation.
Response 1: Done
Point 2: Line no. 22; “(5mg/kg)” There is a space between the number and unit Please check out throughout the manuscript.
Response 2: Done
Point 3: Could you please elaborate, why male broilers were selected? What about female?
Response 3: We just used only male broilers and not male and female broilers in order to have homogenous data.
Point 4: Is there any possible explanation of the correlation between fear and DON? It could be due to nervous system damage or any other reasons. If possible, write 2-3 sentences in discussion section.
Response 4: Line 308 to line 313. Few studies suggested that the stressful effect of Fusarium mycotoxins included DON appeared through the alteration of the brain regional neurochemistry. For example, the inclusion of contaminated grains with Fusarium mycotoxin in the diet of broiler chickens increased concentrations of 5-hydroxytryptamine (5HT) in the pons and in the cortex (Sawmy et al., 2004). Knowing that the 5-HT system take place in the regulation of fear, we suggested that DON could affect the fear level in broilers.
Point 5: In some places, comma is missing. Please check out the whole manuscript.
Response 5: Done
Point 6: Delete this sentence “Thereafter, evaluate the efficacies of DON-detoxifying products on those biomarkers specified.”
Response 6: Done
Reviewer 2 Report
This paper (Manuscript ID: animals-944349) examines the effects of deoxynivalenol contaminated diets on productive, morphological and physiological indicators in broiler chickens. This study contains some original and interesting information and can be accepted for printing in Animals. The title of the article is correct and is consistent with its content. The aim of the paper was formulated clearly and succinctly. The Abstract chapter is written correctly. Statistical analysis is done correctly. The experiment design is done well. The Lab methodology are correct.
I only have a few comments:
L: 56 – Change ‘;’ to ‘.’
L: 119 – Please specify exactly how many birds were analyzed in each group – 12/ treatment?. According to Table 5 – control group – 12 birds; DON (5 mg/kg) – 5 birds, and DON (15 mg/kg) – 7 birds.
L: 92 and L: 175-176 – Remove the dots at the end of the table titles (table is continued).
L: 193-194 – Please add microscopic photos of the small intestine for each group of birds.
L: 199, 214, and 220 – Change ‘p’ to ‘p’ (italics)
L: 203 – Change ‘;);’ to ‘;’
L: 218 – Please complete the table header.
L: 253 – Change ‘[31.32] to ‘[31, 32]’
L: 255 – Change ’[10.22.31]’ to ‘[10, 22, 31]’
L: 274 – Change ‘;’ to ‘,’
L: 302-305 – How to explain the lack of differences in the creatine kinase levels between birds fed control diets and birds fed diet with 15 mg DON/kg.
L: 307-309 - However, no significant differences have been reported between the birds fed control diets and birds fed diet with 5 mg DON/kg, and 15 mg DON/kg (see table 7). How to explain this?
Author Response
Point 1: L: 56 – Change ‘;’ to ‘.’
Response 1: Done
Point 2: L: 119 – Please specify exactly how many birds were analyzed in each group – 12/ treatment? According to Table 5 – control group – 12 birds; DON (5 mg/kg) – 5 birds, and DON (15 mg/kg) – 7 birds.
Response 2: Line 203. In all parameters evaluated we used 12 birds per treatment, however in histological determination, some samples were autolyzed and then we could evaluated this parameter in 12 birds for T1 , 5 birds for T2 and 7 birds for T3.
Point 3: L: 92 and L: 175-176 – Remove the dots at the end of the table titles (table is continued).
Response 3: In “animals” template there are dots at the end of the table titles.
Point 4: L: 193-194 – Please add microscopic photos of the small intestine for each group of birds.
Response 4: We don’t use photos of microscopic determination of small intestine because, there were no significant differences between treatments, and all birds seems to be similar.
Point 5: L: 199, 214, and 220 – Change ‘p’ to ‘p’ (italics)
Response 5: Done
Point 6: L: 203 – Change ‘;);’ to ‘;’
Response 6: Done
Point 7: L: 218 – Please complete the table header.
Response 7: Line 218. Effects of DON contaminated feed (5 and 15 mg/kg) on welfare parameters of broiler chickens
Point 8: L: 253 – Change ‘[31.32] to ‘[31, 32]’
Response 8: Done (line 254)
Point 9: L: 255 – Change ’[10.22.31]’ to ‘[10, 22, 31]’
Response 9: Done (line 256)
Point 10: L: 274 – Change ‘;’ to ‘,’
Response 10: Done
Point 11: L: 302-305 – How to explain the lack of differences in the creatine kinase levels between birds fed control diets and birds fed diet with 15 mg DON/kg.
Response 11: May be due to the highly variability between individuals into the same treatment and therefore the higher standard deviation of the means of different treatments
T1: 9532 ± 126; T2: 4412 ± 3011; T3: 7527 ± 5883.
Point 12: L: 307-309 - However, no significant differences have been reported between the birds fed control diets and birds fed diet with 5 mg DON/kg, and 15 mg DON/kg (see table 7). How to explain this?
Response 12: May be due to the high standard deviation of the means of different treatments:
T1: 169 ± 126; T2: 85 ± 52; T3: 246 ± 212.
Reviewer 3 Report
the rationale for conducting research is unclear.
The toxicity of mycotoxins is generally known. A new and very important element in the publication is bird behavior, the others do not bring any new knowledge.
The authors do not use SI units
comments are highlighted in the text

Author Response
Point 1: as quickly lowered, the temperature was changed
The rapid change in temperature could have had a negative effect on the birds and could have been an experimental factor
Response 1: Line 78-79. Temperature in the first two days was 34 ºC, and then the temperature was gradually decreased 3º C weekly to 21 º C until the end of the experiment.
Point 2: on what basis the addition of mycotoxins to feed was estimated
Response 2: For poultry feedingstuffs, the recommended maximum level for DON is set at 5 mg/kg (European Commission, 2006), as we say in the purpose of the study that we tested those two doses of DON mycotoxin to better understand tolerable DON level in broilers feed, to explore potential damages a lower level might cause, and if it is really in the practice conditions DON could tolerate up 15 mg/kg (as was indicated by biliography).
Point 3: were the chickens tagged? How did the authors know which chickens weighed?
Response 3: Line 111. Chickens were tagged on their wing.
Point 4: did the chickens receive feed before slaughter? Have their access been restricted? If so how long?
Response 4: chickens received the feed ad-libitum and before slaughter, the access to feed has not been restricted.
Point 5: what, more information
Response 5: automatic tissue processor system (Leica,TP1020, Spain)
Point 6: which determined the choice of biochemical blood indicators?
Response 6: basing on the bibliography of previous studies. As DON could affect the physiology and health of chickens, blood biochemistry parameters could be good indicators of effect (Andretta et al., 2012)
Point 7: Why such a time chosen? What it comes from?
Response 7: We just applicate the methodology described by Ghareeb et al., 2014
Point 8: insert SI units (mg/kg)
Response 8: we use the mg/kg unit is the way to express the concentration of DON in feed in litterature. Furthermore, the european comission use ug/kg for mycotoxin concentartion in feed.
Point 9: insert SI units (mg/kg)
Response 9: we use the mg/kg unit is the way to express the concentration of DON in feed in litterature. Furthermore, the european comission use ug/kg for mycotoxin concentartion in feed.
Point 10: < 0.05? Or p≤0.05? Please explain, check
Response 10: p ≤0.05, checked in all tables.
Point 11: insert SI units (mg/kg)
Response 11: we use the mg/kg unit is the way to express the concentration of DON in feed in litterature. Furthermore, the european comission use ug/kg for mycotoxin concentartion in feed.
Point 12: < 0.05? Or p≤0.05? Please explain, check
Response 12: p ≤0.05, checked in all tables.
Point 13: It is a pity that the authors did not control the immune response of the birds
Response 13: The effect of DON on immune response of broiler chickens was presented as another manuscript.
Point 14: insert SI units
Response 14: See table 6 (Total protein : g/L)
Point 15: insert SI units
Response 15: See table 6 (glucose / choleterol / acid uric : mmol/L
Point 16: no title, please provide an appropriate title / table name
Response 16: Line 218. Effects of DON contaminated feed (5 and 15 mg/kg) on welfare parameters of broiler chickens
Point 17: p < 0.05
Response 17: p ≤0.05, checked in all tables.
Point 18: the authors do not provide such results, only the weight
Response 18: Line 264. This result was found in the study of Yunus et al.,2012. In our study, the decrease of villus height was only numerically and did not reach statistical significance.